# AWGN Removal Using Modified Steering Kernel and Image Matching

**Bong-Won Cheon** [1] and **Nam-Ho Kim** [2,*]

1 Department of Intelligent Robot Engineering, Pukyong National University, Busan 48513, Republic of Korea
2 School of Electrical Engineering, Pukyong National University, Busan 48513, Republic of Korea
* Correspondence: nhk@pknu.ac.kr

**Abstract:** Image noise occurs during acquisition and transmission and adversely affects processes, such as image segmentation and object recognition and classification. Various techniques are being studied for noise removal, and with the recent development of hardware and image processing algorithms, noise removal techniques that combine non-local techniques are attracting attention. However, one disadvantage of this method is that blurring occurs in the edges and boundary line of the resulting image. In this study, we proposed a modified local steering kernel based on image matching to improve these shortcomings. The proposed technique uses image matching to differentiate the weight obtained by the steering kernel according to the local characteristics of the image and calculates the weight of the filter based on the similarity between the center window and the matching window. The resulting images were quantitatively evaluation and enlargement of images were used and compared with the existing noise removal algorithms. The proposed algorithm showed clearer contrast in the enlarged images and better results than the conventional image restoration techniques in the quantitative evaluation using peak signal-to-noise ratio and structural similarity index.

**Keywords:** AWGN; modified steering kernel; noise removal; image matching

## 1. Introduction

Noise removal of images is an important preprocessing step in systems that detect objects based on images or use algorithms, such as recognition and tracking. It is often difficult to remove noise in areas with many high-frequency components, such as edges and text in images [1,2]. Various filtering techniques for noise removal to improve image quality and achieve excellent images have been proposed. Noise removal methods, such as Split Bregman-anisotropic total variation denoising (SBATV) and Split Bregman-isotropic total variation denoising (SBITV) [3], have been proven to effectively restore noisy images.

Local steering kernel (LSK) [4,5] is an excellent technique for resolving image noise and uncertainty by estimating the local area structure while preserving the features of the original image. In addition, the nonlocal means (NLMeans) [6] algorithm estimates the original image based on the similarity of two image patches, showing excellent performance and attracting significant attention. However, it is difficult to filter noise in the high-frequency region of the image using typical noise removal techniques as image data and noise are heavily mixed in the said region [7–9].

The Wiener filter became one of the most prominent algorithms in the noise removal field after its proposal by Norbert Wiener. The Wiener filter is fundamentally based on calculating statistical estimates from input signals to generate desired signals. The simple structure and excellent efficiency of Wiener filters enable their use across many studies to improve performance.

The 2D adaptive cuckoo search-based Wiener filter (2DACSWF) [10] was proposed to reduce the noise of the contaminated images in the AWGN algorithm. This algorithm

optimizes the weights of the Wiener filter using the adaptive cuckoo search (ACS) method for estimating the original image.

The split Bregman isotropic total variation denoising (SBITV) [3] algorithm was proposed to improve total variation (TV) denoising. TV denoising is an effective method for restoring original images in images with substantial noise levels. However, the resulting images are prone to the staircase effect and edge component loss. SBITV improved such functions using the split Bregman algorithm.

The fuzzy membership function based modified Gaussian filter (FMGF) [2] was proposed to improve the performance of Gaussian filters when used, as AWGN removal algorithms. FMGF calculates the resulting images by adjusting the weighting of the Gaussian filter according to the fuzzy membership functions. Blurring occurs less in edges and text areas that include the primary data of the image.

This paper proposed a modified steering kernel filter algorithm that utilizes image matching to minimizing smoothing, which occurs during the filtering process of the localization techniques. The proposed algorithm sets the weights by comparing the similarities of the localized regions to improve the local steering kernel method, which uses the directional changes in the pixel values of localized regions. Additionally, the proposed algorithm was applied to image matching procedures for comparing similarities. It determines weights based on the distribution characteristics of the pixel values of the center windows inside the matching areas and matching windows. The size of the center and matching windows were adjusted depending on the noise level of the image using adaptive window sizes. As the mask size was increased, the reduction of noise was increased whereas a smaller mask size pertained the original image details. The proposed algorithm was compared with conventional methods through analysis of simulations and usage of PSNR and SSIM, with the results showing the superior noise reduction of the proposed method compared to the other methods.

## 2. Modified Steering Kernel and Image Matching

When taking an image using a camera and a sensor, noise may be introduced due to a problem in the system or environment. Additive white Gaussian noise (AWGN) [8], a typical noise found in images, is an additive noise evenly distributed throughout the image. The equation of the image in which the AWGN is generated may be expressed as $I_{i,j} = Z_{i,j} + N$. Here, $Z_{i,j}$ denotes an original image not damaged by noise, and $N$ refers to an AWGN with a mean value of 0 and a standard deviation σ. $i, j$ are the internal coordinates of an image with horizontal and vertical dimensions $M \times N$, respectively. The proposed algorithm sets the center window on the image and proceeds with the noise removal process. The center window $W_{i,j}^C(k, l)$ is fixed as a square around the pixel coordinates $(i, j)$. The size of $(2s + 1) \times (2s + 1)$ is set according to parameter $s$, representing the size of the center window. The internal coordinates are set as $k, l$.

The window size used for filtering is a key factor affecting the filtered results. Here, a larger window size reduces noises but introduces a blurring effect that ultimately removes important information such as the edge regions and text components of the image. In contrast, a smaller window size does not reduce noise well. To resolve these drawbacks, an adaptive window size was used to properly size the mask depending on the level of noise.

The adaptive window size $s$ proposed in this study is defined as:

$$s = max[\hat{s}, 1], \ \hat{s} = round[\alpha \cdot \sigma_{est}], \tag{1}$$

where $\alpha$ is the window size parameter, and $\sigma_{est}$ is a noise estimate obtained using noise estimation [11]. The $round[\ ]$ function means rounding and if the noise estimate is low and $\hat{s} = 0$, the window size becomes the lowest value 1.

### 2.1. Local Steering Kernel

The local steering kernel is one of the weights mainly used in image processing and can analyze the slope and direction of the pixel value [12]. The steering kernel regression (SKR) [13,14] is used in calculating the local steering kernel and depends on the pixel position and intensity as well as the unique local structure of the sample. The size and shape of the local steering kernel can extract the structural features of an image and have the effect of spreading the kernel to areas with high correlation with each other.

The local steering kernel $LSK_{x,y}$ is expressed as:

$$LSK_{x,y} = \frac{\sqrt{det(C)}}{2\pi h^2} exp\left\{ -\frac{[p\ q]^T C\ [p\ q]}{2h^2} \right\}, det(C) \geq 0, \tag{2}$$

where

$$x = x_1, x_2, \cdots, x_{2g+1},\ x = i + p,\ \forall\ p \in [-g,\ g], \tag{3}$$

where

$$y = y_1, y_2, \cdots, y_{2g+1},\ y = j + q,\ \forall\ q \in [-g,\ g], \tag{4}$$

where $h$ is the global smoothing parameter, which controls the filter strength. The higher the $h$ value, the stronger the smoothing effect. $x,\ y$ are other pixel coordinates in the local area with respect to the input pixel centered on coordinates $i,\ j$. $g$ is a constant representing the size of the matching area. $[p\ q]$ refers to the $2 \times 1$ matrix, and $p$ and $q$ represent the horizontal and vertical coordinates inside the matching area, respectively. $C$ is a $2 \times 2$ matrix based on the local gradient from a symmetric gradient covariance matrix calculated on a square-shaped local window.

### 2.2. Modified Steering Kernel Weight and Image Matching

The steering kernel has a shape in which the Gaussian weights are inclined according to the gradient characteristics of the local region. As the steering kernel only includes fragmentary information of the local area, smoothing occurs in the resulting image. We used a matching window and image matching to assign large weights to regions with similar pixel distributions.

The matching window $M_{x,y}(k,\ l)$ is set around the pixel coordinates $x, y$ located inside the matching area and is set to $(2s + 1) \times (2s + 1)$ of the same size as the center window to compare similarities between the two areas.

The proposed algorithm performs a similarity comparison on two windows to determine the relationship between the center window and the matching area. The similarity comparison involves comparing pixel values located at the same internal coordinates of two masks with each other. The similarity $d_{x,y}$ of the two masks obtained by comparing the center window and matching window is expressed as:

$$d_{x,y} = \frac{1}{(2s + 1)^2} \sum_{k,l=-s}^{s} \left( C_{x,y}(k,\ l) - M_{x,y}(k,\ l) \right)^2. \tag{5}$$

$k,\ l$ in Equation (5) are discrete variables representing the coordinates inside the window as integers. A lower $d_{x,y}$ value indicates high similarity between the two masks. The proposed algorithm first selects a similarity calculation result that is less than the threshold value and then uses the pixel value of the corresponding coordinate for the final output calculation. The weight $t_{x,y}$ according to the difference and threshold value of the two masks is defined as:

$$t_{x,y} = \hat{d} - \frac{\sqrt{d_{x,y}}}{\sigma_{est}}, \tag{6}$$

Here, $\hat{d}$ represents the threshold. When the value of $d_{x,y}$ is lower, it is presumed that the center and matching windows are similar, and a greater weight is applied. The weight

is computed by comparing the pixel values at the same positions inside the two masks, where $t_{x,y}$ becomes smaller as the pixel value difference between the two masks decreases.

Figure 1 illustrates an example of image matching proposed by this paper. The image used in the example is an 8-bit grey image distorted due to AWGN at $\sigma = 25$. Figure 2 shows a zoomed view of the center of Figure 1 and matching windows A, B, and C.

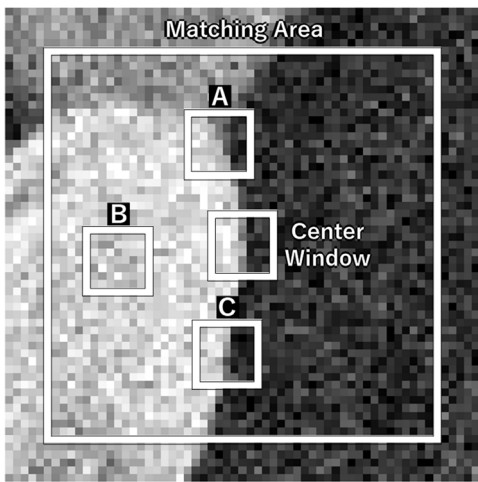

**Figure 1.** Example of image matching on test image.

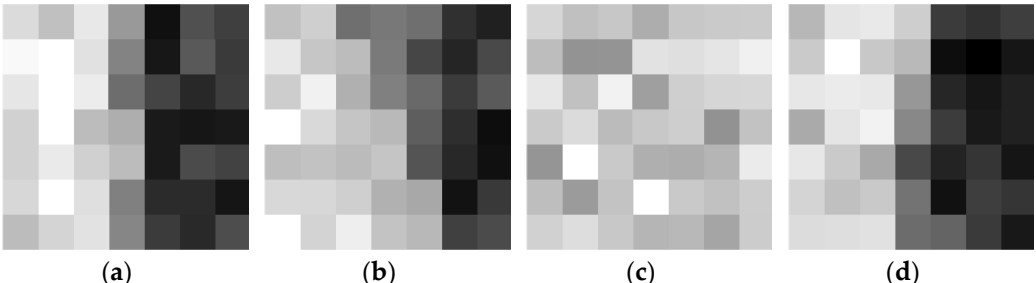

**Figure 2.** Enlarged image of sampling area in test image 1 (**a**) Center window (**b**) Matching window A ($t_{x,y} = 0.1038$) (**c**) Matching window B ($t_{x,y} = -2.3351$) (**d**) Matching window C ($t_{x,y} = 0.5135$).

If the coefficients used for calculation of weights are $\alpha = 0.1$, $h = 1.5$, $g = 10$, and $\hat{d} = 1.5$, the matching window A is a region with similar distribution of pixel values, resulting in $t_{x,y} = 0.1038$. The matching window B is a region with largely different distribution of pixel values, resulting in $t_{x,y} = -2.3351$. The matching window C is a region with the most similar distribution of pixel values to the center window, resulting $t_{x,y} = 0.5135$, which exhibits the largest value among the three matching windows in Figure 1. Similar to the matching window B, the regions with large differences in pixel value distributions between the two windows may have negative weights, and using negative values in result calculations may result in errors. Thus, as shown in the following mathematical expression, the proposed algorithm establishes the weight as 0 when $t_{x,y}$ is negative.

In this case, negative values adversely affect the final value during calculation. Thus, the proposed algorithm uses the following formula to exclude negative values from Equation (6):

$$T_{x,y} = \begin{bmatrix} max\{(t_{1,1}),\ 0\} & max\{(t_{1,2}),\ 0\} & & max\{(t_{1,y}),\ 0\} \\ & & \cdots & \\ max\{(t_{2,1}),\ 0\} & max\{(t_{2,2}),\ 0\} & & max\{(t_{2,y}),\ 0\} \\ & \vdots & \ddots & \vdots \\ max\{(t_{x,1}),\ 0\} & max\{(t_{x,2}),\ 0\} & \cdots & max\{(t_{x,y}),\ 0\} \end{bmatrix}. \tag{7}$$

To further emphasize area of higher similarity, the weights given differentially according to the threshold are applied to the local steering kernel $\omega_{x,y}^{LSK}$. The modified steering kernel weights $U_{x,y}$ calculated based on the two weights is as follows:

$$U_{x,y} = \begin{bmatrix} T_{1,1}\ \omega_{1,1}^{LSK} & T_{1,2}\ \omega_{1,2}^{LSK} & & T_{1,y}\ \omega_{1,y}^{LSK} \\ & & \cdots & \\ T_{2,1}\ \omega_{2,1}^{LSK} & T_{2,2}\ \omega_{2,2}^{LSK} & & T_{2,y}\ \omega_{2,y}^{LSK} \\ & \vdots & \ddots & \vdots \\ T_{x,1}\ \omega_{x,1}^{LSK} & T_{x,2}\ \omega_{x,2}^{LSK} & \cdots & T_{x,y}\ \omega_{x,y}^{LSK} \end{bmatrix}. \tag{8}$$

The filtering resultant image $\hat{Z}_{i,j}$ obtained from Equation (8) is expressed as:

$$\hat{Z}_{i,j} = \frac{1}{u} \sum_{x,y=-g}^{g} U_{x,y} I_{x,y}, \tag{9}$$

$x, y$ in Equation (9) are discrete variables representing the coordinates inside the matching area as integers. $u$ denotes a normalizing parameter of the weight $U_{x,y}$.

Figure 3 shows the flowchart of the proposed algorithm. The flowchart shows the modified steering kernel weight setting, image matching, and filter output calculation process of the proposed algorithm.

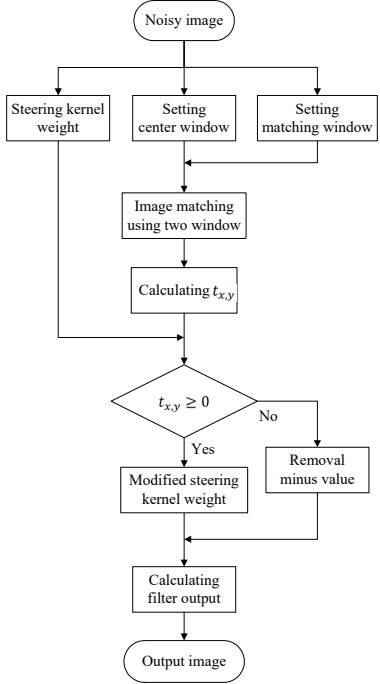

**Figure 3.** Flowchart of proposed filter algorithm.

### 3. Simulation and Results

#### 3.1. Experimental Setting

A series of experiments were performed to verify the effectiveness of the proposed algorithm. The noise removal function was objectively evaluated using $512 \times 512$ 8-bit gray images in the simulation, as shown in Figure 4. The standard deviation of the AWGN, ranging from 5–30, was prepared to evaluate and analyze the denoising performance based on the noise level of the proposed algorithm. Figure 5 shows the four types of test images used in the simulation and the noise image corrupted by the AWGN with $\sigma = 30$.

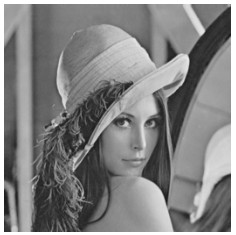 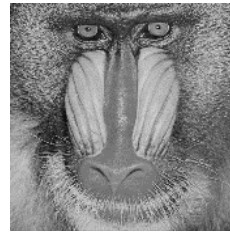 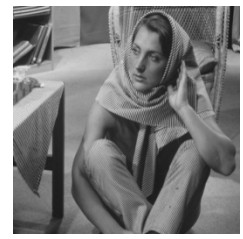 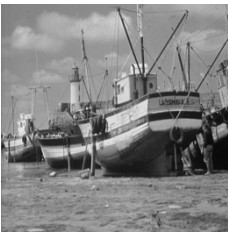

**Figure 4.** Original image of test images (Lena, Baboon, Barbara, and Boat).

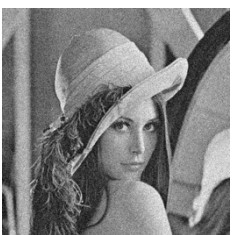 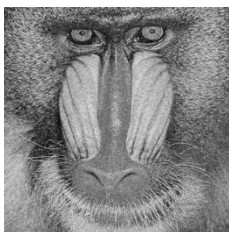 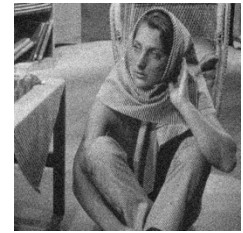 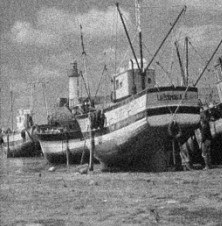

**Figure 5.** Corresponding noisy images corrupted by AWGN ($\sigma = 30$).

The parameters used in the proposed algorithm were simulated using multiple test images. After the resulting images were analyzed according to the changes in the parameter values, the parameters were selected such that excellent PSNR and SSIM [15–18] characteristics could be achieved in images other than the test images. Table 1 shows the optimal values of the parameters used in the proposed algorithm.

**Table 1.** Parameter set of proposed modified steering kernel filter.

| Parameter | Variable | Value |
|---|---|---|
| Window size parameter | $\alpha$ | 0.1 |
| Smoothing parameter | $h$ | 1.5 |
| Matching area size | $g$ | 10 |
| Filter weight threshold | $\hat{d}$ | 1.5 |

The higher the value of the window size parameter, the higher the size of the center and matching windows, thus leading to an enhanced AWGN-removal function. However, exceedingly high values lead to more intense smoothing effects, and the edge component becomes vague. The filter weight threshold is a constant that determines the number of matching windows that has been used to calculate weights through window matching. The larger the filter weight threshold, the more the matching windows are used to calculate weights; however, a result image obtained with more matching windows can be blurred as it may include values lacking relevance. The smoothing parameter and matching area size were set by referring to the values in [6] and [3], respectively. The smoothing parameter is a constant that determines the shape of the steering kernel, and the smaller its value, the higher the weight set in the center area of the kernel. A high smoothing parameter can distribute the weight throughout the kernel. The matching area size is a constant that

determines the area where window matching progresses; the higher its value, the more the areas that can be matched, thus leading to improved AWGN removal performance. However, since the number of matching windows used in this method increases, processing can take longer, or areas with low relevance might be included.

### 3.2. Experimental Result and Comparison

Figures 6–9 show the filtered images and the enlarged portions of the images.

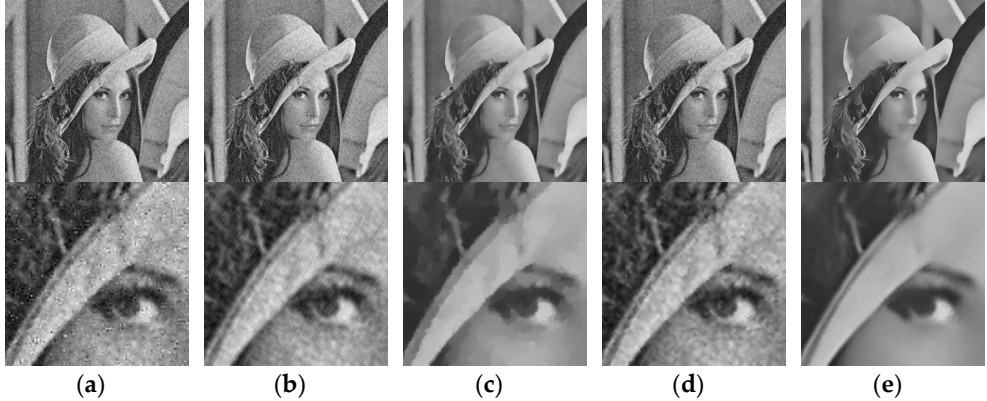

**Figure 6.** De-noising results for Lena image with AWGN of σ = 30 (**a**) WF (**b**) 2DACSWF (**c**) SBITV (**d**) FMGF (**e**) Proposed method.

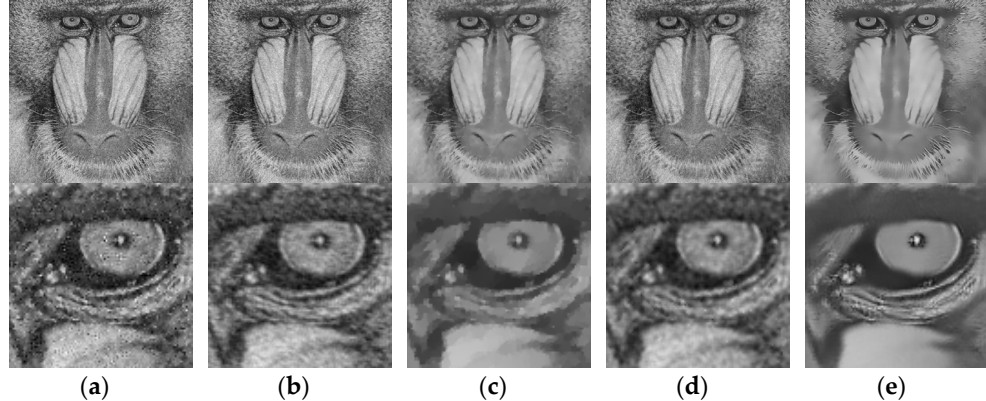

**Figure 7.** De-noising results for Baboon image with AWGN of σ = 30 (**a**) WF (**b**) 2DACSWF (**c**) SBITV (**d**) FMGF (**e**) Proposed method.

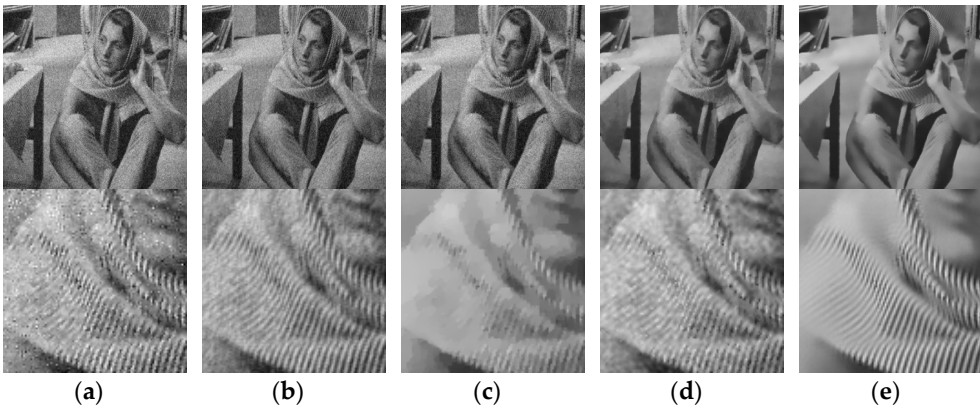

**Figure 8.** De-noising results for Barbara image with AWGN of σ = 30 (**a**) WF (**b**) 2DACSWF (**c**) SBITV (**d**) FMGF (**e**) Proposed method.

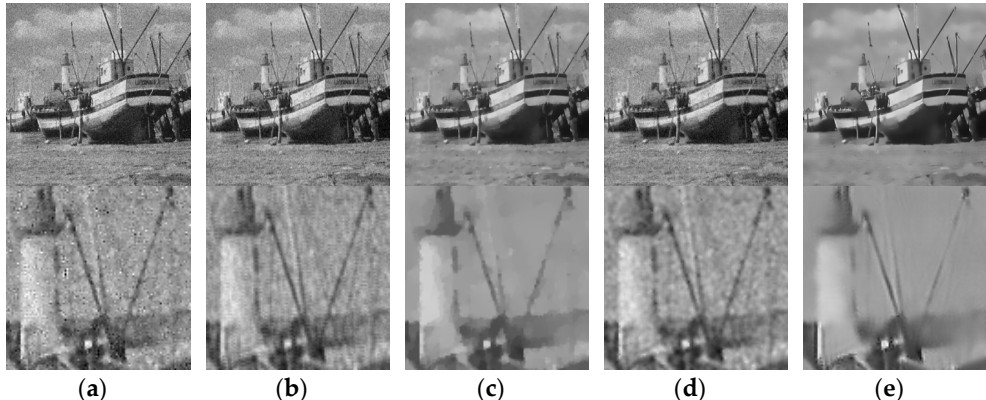

|  |  |  |  |  |
|---|---|---|---|---|
| (**a**) | (**b**) | (**c**) | (**d**) | (**e**) |

**Figure 9.** De-noising results for Boat image with AWGN of $\sigma = 30$ (**a**) WF (**b**) 2DACSWF (**c**) SBITV (**d**) FMGF (**e**) Proposed method.

The resulting images of conventional noise removal methods and the proposed algorithm were compared to observe the visual effects.

### 3.3. Comparison of PSNR and SSIM Results

The performance of the proposed algorithm was quantitatively evaluated using the peak signal-to-noise ratio (PSNR) and structure similarity index (SSIM). Table 2 shows the result of the proposed algorithm and other existing methods on distorted images with AWGN values of 5–30. Figure 10 shows a graphical representation of Table 2.

**Table 2.** Comparison of PSNR and SSIM.

| Image | AWGN [$\sigma$] | WF | | 2DACSWF | | SBITV | | FMGF | | PFA | |
|---|---|---|---|---|---|---|---|---|---|---|---|
| | | PSNR [dB] | SSIM | PSNR [dB] | SSIM | PSNR [dB] | SSIM | PSNR [dB] | SSIM | PSNR [dB] | SSIM |
| Lena | 5 | 37.33 | 0.9280 | 32.57 | 0.7996 | 36.54 | 0.9224 | 34.97 | 0.9189 | 37.97 | 0.9380 |
| | 10 | 33.25 | 0.8498 | 30.15 | 0.6393 | 33.62 | 0.8832 | 33.23 | 0.8619 | 34.54 | 0.8952 |
| | 15 | 30.69 | 0.7631 | 28.44 | 0.6460 | 31.90 | 0.8554 | 31.29 | 0.7876 | 32.84 | 0.8684 |
| | 20 | 28.76 | 0.6767 | 26.21 | 0.5331 | 30.76 | 0.8332 | 29.49 | 0.7056 | 31.50 | 0.8418 |
| | 25 | 27.22 | 0.6017 | 25.71 | 0.5417 | 29.94 | 0.8170 | 27.96 | 0.6330 | 30.69 | 0.8197 |
| | 30 | 26.00 | 0.5379 | 25.08 | 0.4863 | 29.18 | 0.7994 | 26.64 | 0.5650 | 29.74 | 0.7980 |
| Baboon | 5 | 33.27 | 0.9108 | 29.08 | 0.8712 | 31.01 | 0.9145 | 26.19 | 0.6779 | 33.84 | 0.9148 |
| | 10 | 29.55 | 0.8615 | 26.89 | 0.7745 | 27.02 | 0.8128 | 25.73 | 0.6019 | 30.04 | 0.8700 |
| | 15 | 27.13 | 0.7204 | 24.46 | 0.6612 | 25.03 | 0.7244 | 25.19 | 0.5360 | 27.42 | 0.7778 |
| | 20 | 25.41 | 0.7148 | 23.90 | 0.6156 | 23.84 | 0.6536 | 24.54 | 0.4806 | 25.89 | 0.7201 |
| | 25 | 24.11 | 0.6126 | 22.97 | 0.5916 | 23.00 | 0.5958 | 23.83 | 0.4311 | 24.79 | 0.6264 |
| | 30 | 23.07 | 0.5328 | 22.87 | 0.5492 | 22.39 | 0.5470 | 23.08 | 0.3893 | 23.99 | 0.5611 |
| Barbara | 5 | 36.17 | 0.9442 | 30.98 | 0.7274 | 32.84 | 0.9230 | 27.19 | 0.8383 | 37.37 | 0.9527 |
| | 10 | 31.50 | 0.8780 | 27.05 | 0.5529 | 28.74 | 0.8438 | 26.80 | 0.7977 | 33.29 | 0.9196 |
| | 15 | 28.72 | 0.7930 | 26.85 | 0.5289 | 26.79 | 0.7803 | 26.26 | 0.7423 | 31.19 | 0.8911 |
| | 20 | 26.83 | 0.7100 | 25.03 | 0.4575 | 25.71 | 0.7350 | 25.59 | 0.6809 | 29.52 | 0.8545 |
| | 25 | 25.35 | 0.6330 | 24.62 | 0.4285 | 24.98 | 0.7019 | 24.81 | 0.6179 | 28.49 | 0.8193 |
| | 30 | 24.21 | 0.5669 | 23.67 | 0.3591 | 24.47 | 0.6761 | 24.07 | 0.5603 | 27.39 | 0.7809 |
| Boat | 5 | 35.62 | 0.9061 | 28.53 | 0.8394 | 34.50 | 0.8946 | 31.78 | 0.8747 | 36.28 | 0.9137 |
| | 10 | 32.06 | 0.8420 | 28.18 | 0.6891 | 31.44 | 0.8348 | 30.80 | 0.8342 | 33.01 | 0.8694 |
| | 15 | 29.68 | 0.7682 | 27.24 | 0.6827 | 29.73 | 0.7905 | 29.57 | 0.7790 | 30.84 | 0.8066 |
| | 20 | 27.92 | 0.6961 | 24.61 | 0.5753 | 28.59 | 0.7566 | 28.18 | 0.7139 | 29.55 | 0.7669 |
| | 25 | 26.45 | 0.6288 | 24.42 | 0.5352 | 27.70 | 0.7292 | 26.91 | 0.6508 | 28.58 | 0.7155 |
| | 30 | 25.27 | 0.5685 | 23.90 | 0.5037 | 26.98 | 0.7055 | 25.78 | 0.5916 | 27.74 | 0.6849 |

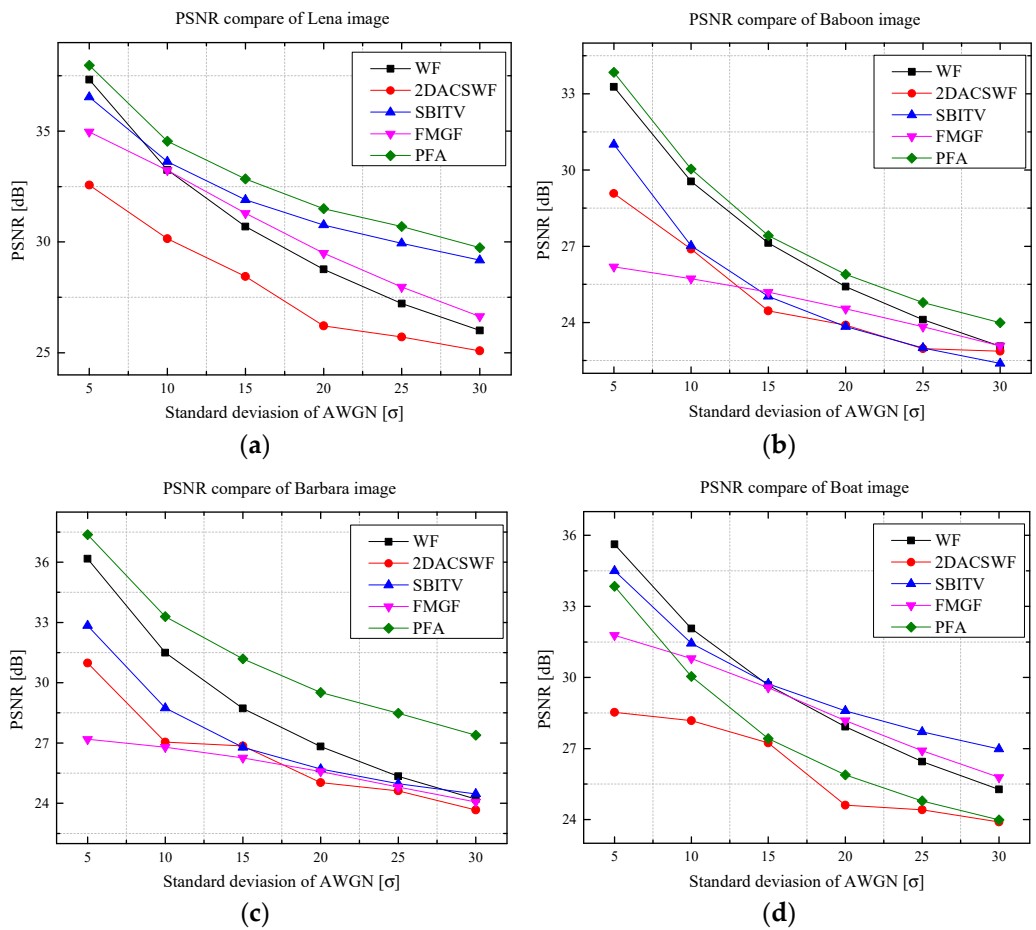

**Figure 10.** PSNR graph of simulation result. (**a**) PSNR graph of Lena image. (**b**) PSNR graph of Baboon image. (**c**) PSNR graph of Barbara image. (**d**) PSNR graph of Boat image.

*3.4. Performance Comparisons*

Visual and quantitative evaluations were conducted to assess the performance of the proposed algorithm. Simulations were carried out using MATLAB R2020b software running on an Intel Core i7-10700 system with 2.90-GHz CPU, 16-GB RAM, and a 64-bit operating system.

For visual evaluation, certain regions from the resulting images were enlarged as shown in Figures 6–9. The resulting images processed using WF, 2DACS-WF, and F-MGF were not completely free of noise and were heavily affected by the AWGN. The results from the SB-ITV exhibited damages at the intersections between bright and dark regions, indicating that edge characterization was difficult compared to other methods. Conversely, the results from the proposed algorithm clearly maintained the edge components while significantly reducing the level of noise compared to other methods.

In Table 2, the PSNR and SSIM of the proposed algorithm were superior to the conventional methods. In the case of the distorted Barbara image at AWGN $\sigma = 30$, the proposed algorithm showed a PSNR value of 27.39 [dB]. The proposed algorithm's PSNR was better than WF, 2DACSWF, SBITV, and FMGF by 3.18 [dB], 3.72 [dB], 2.92 [dB], and 3.32 [dB], respectively.

Concerning the proposed algorithm, the PSRN and SSIM results show high or similar values in most areas when restoring a noisy image from $\sigma = 5$ when the intensity of the AWGN is relatively weak to $\sigma = 30$, which indicates prominent noise. In particular, the proposed algorithm shows a significant performance increase compared with conventional methods when restoring images with high-frequency components, such as the Barbara images. Table 1 shows that the proposed algorithm had a PSNR of 27.39 [dB], which was

higher than the PSNR values obtained using WF, SBITV, and FMGF processing by 3.18 [dB], 3.72 [dB], 2.92 [dB], and 3.32 [dB], respectively, by restoring the Barbara image damaged using an AWGN of $\sigma = 30$. In summary, the visual and quantitative results show that the proposed algorithm effectively eliminates AWGN and inhibits excessive blurring of edge components.

## 4. Conclusions

In this study, we propose a modified steering kernel algorithm based on image matching to improve the blurring phenomenon that occurs in the edges of the image as a result of AWGN removal. As a typical steering kernel only contains gradient information for a local area of the image, blurring occurs while filtering high-frequency components, such as edges. To solve this problem, the proposed algorithm proposes a modified steering kernel algorithm based on image matching.

The proposed algorithm determined the modified steering kernel weight by utilizing the weights derived from image matching applied to the original steering kernel. The image matching was processed by utilizing the similarity between center and matching windows, and the final images were computed using the modified steering kernel weights applied to the pixel values of the matching areas. The performance of the proposed algorithm was evaluated using the resulting and enlarged images. According to the simulation, the proposed algorithm displayed better edge components, a higher contrast in the high frequency areas, and a larger reduction of noise by suppressing the blurring effect. The PSNR and SSIM were used for more quantitative evaluation, with the proposed algorithm displaying superior results compared to existing methods as a result of comparing different types of images and various levels of noise.

The proposed algorithm has excellent noise removal functions but requires a relatively long time to eliminate noise owing to the increased computational complexity. In the future, we intend to conduct research aiming to simplify and optimize the algorithm to solve these shortcomings and enhance the overall performance.

**Author Contributions:** Conceptualization, B.-W.C. and N.-H.K.; Software, B.-W.C.; Validation, N.-H.K.; Formal analysis, B.-W.C.; Investigation, B.-W.C.; Data curation, B.-W.C.; Writing—original draft preparation, B.-W.C.; Writing—review and editing, B.-W.C. and N.-H.K.; Visualization, B.-W.C.; project administration, N.-H.K.; All authors have read and agreed to the published version of the manuscript.

**Funding:** This research received no external funding.

**Institutional Review Board Statement:** Not applicable.

**Informed Consent Statement:** Not applicable.

**Data Availability Statement:** Not applicable.

**Conflicts of Interest:** The authors declare no conflict of interest.

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
