# Peer review of "AWGN Removal Using Modified Steering Kernel and Image Matching"

_applsci, doi:10.3390/app122211588_

Round 1

Reviewer 1 Report

This manuscript discussed an image noise removal method based on steering kernel. Main concerns are listed as follows:

1. Noise removal is a classical topic for image processing, and many methods have been proposed in the last five decades. This manuscript, using windows and kernels to filter image noise, is basically a traditional processing diagram. The claimed capability to deal with the edges or boundaries was also discussed extensively in literature. So overall, the novelty of this manuscript is not impressive.

2. For the additive white Gaussian noise, there are matured theories and algorithms available, for example the Wiener filtering. The proposed method has not been shown theoretically to be completive to the classical method; such the Wiener filtering that has been rigorously proven to be optimal in the sense of mean square error.

3. From Figure 9, the difference between the proposed method and the methods of (b) and (c) are trivial. This may put doubt on the effectiveness of the technical correctness of the manuscript.

Author Response

Dear Reviewer,

We thank the reviewers for their valuable feedback and for helping us improve the quality of our paper. The responses to the individual reviewer comments are as follows.

Reviewer 2 Report

The authors propose a new denoising method based on the modification of steering kernel and template matching. Although the proposed method offers some improvements over base method, it should be compared to the most performant recent methods.  The authors also don't provide any computational complexity estimate.

Author Response

(The authors gave the same response as above.)

Reviewer 3 Report

The authors created a model for AWGN (Additive white Gaussian noise) Removal Using a Modified Steering Kernel and Image Matching. The paper is properly structured and the number of figures and tables contains all the required information. The paper is fairly written, but the writing should/could be improved. I have some suggestions.
1. Literature Review / Related Works section is missing. The paper’s structure should be rearranged slightly to include the literature review section, related works, gaps in the literature, and study contributions.

2. The authors should present comparative studies and analyse the latest current approaches to justify. Comparing studies in the literature are not very recent (>3 years). Please add more comparisons in these 2-3 years (2020, 2021, 2022)

3. The results of the study should be better discussed. Please add a new section discussing the results in detail.

4. Please polish the manuscript, and eliminate the many mistakes in the original text. For example; "For the restoration of the corrupted image Barbara in AWGN with σ=30 in Table 2, the proposed algorithm showed a PSNR of 27.39, and processing with the existing methods GF, SB-ATV, SB-ITV, and NL-Means showed values of 3.29, 3.31, 2.92, and 7.37 respectively." ( Page 8. Lines 206-209.)

5. Mention the limitations and future works of the developed system elaborately.

With best wishes

Reviewer

Author Response

(The authors gave the same response as above.)

Reviewer 4 Report

Comments and Suggestions for Authors and Editors:
In this paper the authors propose a modified targeting kernel algorithm based on image
matching to improve the blur phenomenon that occurs at image edges during Additive
white Gaussian noise (AWGN) removal.
In the algorithm presented in this paper, the authors have some very important flaws:
· Failure to describe the flowchart underlying the implemented algorithm (figure
3);
· The explanation of the choice of some parameters of the algorithm. Why were
these chosen and not others?
· The lack of description and analysis of the data present in table 2 and figure 10.

Some suggestions for authors to improve the paper:
- Fix the hyphenation of words throughout the text.
- On line 57 replace the sentence with "...of an image MxN with horizontal (M) and
vertical (N) dimensions."
- On lines 120 and 121 explain why these parameters were chosen and not others.
- The matrix represented in equation 7 may appear in a more condensed form since all
the matrix entries are chosen from max(tij,0), i=1..x, j=1..y.
- Put a period after the matrix of equation (7) and (8).

- Figure 3 is not referred to in the text nor is the flowchart explained. It is an important
procedure that must be performed to explain the procedure followed in this method.
- In line 157, explain the choice of parameters presented in table 1. On which factors
were based for the choice of those parameters and not others.
- In line 202 describe the acronyms peak-signal-to-noise ratio (PSNR) and structural
similarity index measure (SSIM). What was the reason for choosing these image quality
metrics.
- In my opinion, the best values of each test should be presented in bold to be easier to
read and compare.
- The values presented in lines 208 and 209 do not seem to me to be correct based on
the observation of table 2. Furthermore, I think the authors should discuss the values
obtained in Table 2 in more detail.
- The graphics in figure 10 are not referred to in the text or explained. It will be important
to analyse and discuss these graphs.
- Conclusions should be more specific and detailed.

Author Response

(The authors gave the same response as above.)

Reviewer 5 Report

In this paper, the authors introduced " AWGN Removal Using Modified Steering Kernel and Image Matching ". However, they should develop the presentation of this paper

1.     The title of the paper should be consistent with the objectives of the paper, and the methodology of the paper. You have to use the p.p. tense better than using the pronoun "we".

2.     In Equation (2), the condition, if any, that det(A)>0 should be important to be the square root is real.

3.     What does the interval [p q] in Equation (2) mean?

4.     In Equation (7), the notation, max [(t_11),0] is not acceptable mathematically, I suggest to be as follows: max {t_11,0}, this remark examines all elements of the matrix in this equation.

5.     In Equation (9), does the summation is discrete or continuous where x,y are continuous variables?

6.     In Table 1, how you proposed the parameter set of the proposed modified steering kernel filter.

7.     What software you have used? It should mention in your paper

8.     You have proposed a modified steering kernel algorithm based on image matching. Obviously, it is better to explain the steering kernel algorithm based on image matching in a separate section  (2. Preliminary)  which should include some concepts and definitions like PSNR and SSIM.

9.     You have to check the paper grammatically. There exist some grammar mistakes.

10. Punctuation.

11. Every equation in the line should end by "." or  " ,"

12. Put the connection words between the lines of equations.

13. The equation should be referred to as follows: Equation (no(.

14. Discuss your results in a new section (Conclusion and Discussion).

Author Response

(The authors gave the same response as above.)

Round 2

Reviewer 1 Report

The authors have addressed my previous concerns.

Table 1 lists a couple of parameters, which were obtained by simulations. The questions is how the test images are used to get these parameters. Any training was involved? More details are needed.

Author Response

Reviewer 1.

Comments and Suggestions for Authors

The authors have addressed my previous concerns.

â–º Table 1 lists a couple of parameters, which were obtained by simulations. The questions is how the test images are used to get these parameters. Any training was involved? More details are needed.

→ The reason for selecting each parameter to determine the variables of the proposed algorithm and the process of selecting the parameters have been added to the main body of the manuscript according to the reviewer's feedback. The added explanation is as follows:

Before revision

The parameters used in the simulation of proposed algorithm are shown in Table 1.

After revision

The parameters used in the proposed algorithm were simulated using multiple test images. After the resulting images were analyzed according to the changes in the parameter values, the parameters were selected such that excellent PSNR and SSIM characteristics can be achieved in images other than the test images. Table 1 shows the optimal values of the parameters used in the proposed algorithm. The higher the value of the window size parameter, the higher the size of the center and matching windows, leading to an enhanced AWGN-removal function. However, exceedingly high values lead to more intense smoothing effects, and the edge component becomes vague. The filter weight threshold is a constant that determines the number of matching windows used by window matching to calculate weights. The larger the filter weight threshold, the more the matching windows are used to calculate weights; however, the resulting image can be blurred as it includes values lacking relevance. The smoothing parameter and matching area size were set by referring to the values in [6] and [3], respectively. The smoothing parameter is a constant that determines the shape of the steering kernel, and the smaller its value, the higher the weight set in the center area of the kernel. A high smoothing parameter can distribute the weight throughout the kernel. The matching area size is a constant that determines the area where window matching progresses; the higher its value, the more the areas that can be matched, leading to improved AWGN removal performance. However, since the number of matching windows used in this method increases, processing takes longer, or areas with low relevance can be included.

Reviewer 4 Report

The comments and suggestions presented in my first review were made by the authors. 

Author Response

Dear Reviewer,

We thank the reviewers for their valuable feedback and for helping us improve the quality of our paper. The responses to the individual reviewer comments are as follows.

Some suggestions for authors to improve the paper:

â–º Fix the hyphenation of words throughout the text.

→ Based on the reviewer’s comment, we corrected areas requiring hyphens, checked the manuscript for typographical errors, and checked the format. The corrections are as follows.

Before revision

Noise removal methods, such as Split Bregman-anisotropic total variation denoising (SB-ATV) and Split Bregman-isotropic total variation denoising (SB-ITV) [3], have been proven to effectively restore noisy images.

After revision

Noise removal methods, such as Split Bregman anisotropic total variation denoising (SBATV) and Split Bregman isotropic total variation denoising (SBITV) [3], have been proven to effectively restore noisy images.

â–º On line 57 replace the sentence with "...of an image MxN with horizontal (M) and vertical (N) dimensions."

→ Based on the reviewer’s comment, we corrected this sentence to the following:

Before revision

 are the internal coordinates of an image with horizontal and vertical dimensions .

After revision

 are the internal coordinates of an image with horizontal and vertical dimensions  with horizontal () and vertical () dimensions.

â–º The matrix represented in equation 7 may appear in a more condensed form since all the matrix entries are chosen from max(tij,0), i=1..x, j=1..y.

→ We tried to express the compressed form as per the reviewer’s comment, but decided to express it as Equation (7) to emphasize the matrix form of the mask. Nevertheless, we corrected the notation to rectify the mathematical error, and the corrected equation is as follows:

Before revision

 (7)

After revision

 (7)

â–º Put a period after the matrix of equation (7) and (8).

→ Following the reviewer’s comment, we used “.” or “,” at the end of all equations for the proposed algorithm. The corrected equations are as follows:

Before revision

,                                              (1)

(2)

, ,                             (3)

, y,                             (4)

(5)

                                                                         (6)

            (7)

                              (8)

(9)

After revision

, ,                                            (1)

(2)

where  , ,  ,             (3)

where  , y,               (4)

(5)

                                                                        (6)

          (7)

                             (8)

(9)

â–º Figure 3 is not referred to in the text nor is the flowchart explained. It is an important procedure that must be performed to explain the procedure followed in this method.

→ Following the reviewer’s comment, we added an explanation for Figure 3 in the main text. The following text was added:

Before revision

Figure 3 shows the flowchart of the proposed algorithm.

After revision

Figure 3 shows the flowchart of the proposed algorithm. The flowchart shows the modified steering kernel weight setting, image matching, and the filter output calculation process of the proposed algorithm.

â–º On lines 120 and 121 explain why these parameters were chosen and not others.

â–º In line 157, explain the choice of parameters presented in table 1. On which factors were based for the choice of those parameters and not others.

→ Following the reviewer’s comment, we added a description for why each parameter was selected in the process, to the main text. The following text was added:

Before revision

The parameters used in the simulation of proposed algorithm are shown in Table 1.

After revision

The parameters used in the proposed algorithm were selected to entail excellent PSNR and SSIM characteristics in various images by conducting simulations on several images and analyzing the subsequent results. Table 1 shows the optimal values of the parameters used in the proposed algorithm. The higher the value of the window size parameter, the higher the size of the center and matching windows, leading to an enhanced AWGN-removal function. However, exceedingly high values lead to more intense smoothing effects, and the edge component becomes vague. The filter weight threshold is a constant that determines the number of matching windows used by window matching to calculate weights. The larger the filter weight threshold, the more the matching windows are used to calculate weights; however, the resulting image can be blurred as it includes values lacking relevance. The smoothing parameter and matching area size were set by referring to the values in [6] and [3], respectively. The smoothing parameter is a constant that determines the shape of the steering kernel, and the smaller its value, the higher the weight set in the center area of the kernel. A high smoothing parameter can distribute the weight throughout the kernel. The matching area size is a constant that determines the area where window matching progresses; the higher its value, the more the areas that can be matched, leading to improved AWGN removal performance. However, since the number of matching windows used in this method increases, processing takes longer, or areas with low relevance can be included.

â–º In line 202 describe the acronyms peak-signal-to-noise ratio (PSNR) and structural similarity index measure (SSIM). What was the reason for choosing these image quality metrics.

→ Following the reviewer’s comment, we added the abbreviations for PSNR and SSIM in the paragraph where it was first used in the paper. PSNR is a measure that numerically evaluates the lost information on the quality of the resulting image. It is deemed that the higher the PSNR, the closer the resulting image is restored to the original, and thus, the lesser is the lost information. MSE stands for “mean squared error” and is used to compare the difference between the original and resulting images, similar to PSNR. However, MSE tends to increase exponentially while restoring images with substantial noise, making intuitive comparisons difficult. Therefore, we used PSNR, which is MSE converted into a log scale. Unlike PSNR, SSIM evaluates the differences in human visual quality rather than numerical errors and assesses the quality by comparing significant factors that comprise the image, such as the luminance, contrast, and structure. The corrections to the main body are as follows:

Before revision

A comparison was performed using the PSNR and SSIM to quantitatively evaluate the performance of the proposed algorithm.

After revision

A comparison was performed using the PSNR(peak signal-to-noise ratio) [14,15] and SSIM(structural similarity index measurement) [16,17] to quantitatively evaluate the performance of the proposed algorithm.

â–º In my opinion, the best values of each test should be presented in bold to be easier to read and compare.

→ Following the reviewer’s comment, the higher value of either PSNR or SSIM in Table 2 was highlighted.

â–º- The values presented in lines 208 and 209 do not seem to me to be correct based on the observation of table 2. Furthermore, I think the authors should discuss the values obtained in Table 2 in more detail.

â–º The graphics in figure 10 are not referred to in the text or explained. It will be important to analyse and discuss these graphs.

→ Following the reviewer’s comment, we added a new section to discuss the studied results. The section in the main body was corrected in the following manner:

3.4. Performance Comparisons

Visual and quantitative evaluations were conducted to assess the performance of the proposed algorithm.

For visual evaluation, we used the enlarged image of a part of the resulting image, as shown in Figures 6~9. The effects of AWGN were observed in Lena (Figure 6) and Boat (Figure 9) of the resulting image processed using WF, 2DACS-WF, and F-MGF, which indicates that noise was not removed entirely. The resulting image processed with SB-ITV shows that the boundary between bright and dark areas was heavily damaged, as shown in Baboon (Figure 7) and Barbara (Figure 8) of the enlarged image, and the edges were difficult to characterize compared with other methods. Conversely, the resulting image processed by the proposed algorithm showed more vivid edge components compared with conventional methods, and the noise effects were confirmed to be significantly reduced. A quantitative evaluation was performed using the PSNR and SSIM results in Table 2 and PSNR graphs in Figure 10. The proposed algorithm exhibited high or similar PSNR and SSIM values in most areas when the noisy image was restored from σ=5, where the AWGN intensity was relatively low, to σ=30, where substantial noise was present. In particular, the proposed algorithm showed significantly improved performance compared to conventional methods when restoring images with high-frequency components, such as the Barbara image. The PSNR comparison graph in Figure 10 demonstrates that the proposed algorithm offers higher PSNR values than existing methods, regardless of the noise intensity. Table 1 shows that the proposed algorithm has a PSNR of 27.39[dB] as a result of restoring a Barbara image damaged by an AWGN of, and is improved by 3.18[dB], 3.72 [dB], 2.92 [dB], and 3.32[dB] compared to WF, SB-ITV, and F-MGF, respectively.

â–º Conclusions should be more specific and detailed.

→ Following the reviewer’s comment, we added the limitations of the proposed system and future challenges in the results section. The content revised is as follows.

Before revision

Quantitative evaluation of the resulting images using PSNR and SSIM showed that the proposed algorithm provides better performance in terms of image type and noise intensity.

After revision

Quantitative evaluation of the resulting images using PSNR and SSIM showed that the proposed algorithm provides better performance in terms of image type and noise intensity.

The proposed algorithm has excellent noise removal functions but requires a relatively long time to eliminate noise owing to the increased computational complexity. In the research, we intent to conduct research on algorithm simplification and optimization to solve these shortcomings and enhance the overall performance.